# Spatiotemporal distributions of under-five mortality in Ethiopia between 2000 and 2019

**Kendalem Asmare Atalell** [1]\*, **Kefyalew Addis Alene**[2,3]

**1** Department of Pediatrics and Child Health Nursing, School of Nursing, College of Medicine and Health Sciences, University of Gondar, Gondar, Ethiopia, **2** Telethon Kids Institute, Nedlands, Western Australia, Australia, **3** Faculty of Health Sciences, Curtin University, Bentley, Western Australia, Australia

\* kedasmar@gmail.com

**Data Availability Statement:** All data are in the manuscript and Supporting information files.

**Funding:** The authors received no specific funding for this work.

## Abstract

Under-five mortality declined in the last two decades in Ethiopia, but sub-national and local progress remains unclear. This study aimed to investigate the spatiotemporal distributions and ecological level factors of under-five mortality in Ethiopia. Data on under-five mortality were obtained from five different Ethiopian Demographic and Health Surveys (EDHS), conducted in 2000, 2005, 2011, 2016, and 2019. Environmental and healthcare access data were obtained from different publicly available sources. Bayesian geostatistical models were used to predict and visualize spatial risks for under-five mortality. The national under-five mortality rate in Ethiopia declined from 121 per 1000 live births in 2000 to 59 per 1000 live births in 2019. Spatial variation in under-five mortality was observed at regional and local levels with the highest rates reported in the Western, Eastern, and Central parts of Ethiopia. Spatial clustering of under-five mortality was significantly associated with population density, access to a water body, and climatic factors such as temperature. Under-five mortality rate declined over the past two decades and varied substantially at sub-national and local levels in Ethiopia. Increasing access to water and health care may help to reduce under-five mortality in high-risk areas. Therefore, interventions targeted to reduce under-five mortality should be strengthened in the areas that had a clustering of under-five mortality in Ethiopia by increasing access to quality health care access.

## Introduction

Under-five mortality is a health indicator and critical measure of human development. Globally, remarkable efforts had been made to reduce under-five mortality for the past few decades. These efforts substantially reduced the annual under-five mortality rate by 3.7% from 2000 to 2019 [1]. Despite this progress, an estimated 5.2 million children under the age of 5 years died in 2019 [2, 3], and almost 14,000 children died every day.

The burden of under-five mortality varies at regional and national levels [3] with the highest death reported in Africa (74 per 1000 live births) [4, 5], which accounted for more than half of the global burden [4]. In 2019, more than 80% of the global under-five death occurred in sub-Saharan Africa [6–8].

**Competing interests:** The authors have declared that no competing interests exist.

The Sustainable Development Goals (SDGs) aimed to end all preventable deaths of under-5 children by 2030, targeted to reduce under-five mortality below 25 per 1,000 live births in every country [9–11]. Great efforts have been made to achieve this target by increasing access and providing quality maternal and child health services [12, 13]. These efforts have prevented childhood illness and improved the survival of children. However, under-five mortality is still a major problem in developing countries such as Ethiopia [14]. Ethiopia is among the five countries which share greater than half of the world's under-five death [1, 2]. According to the UN Inter-agency Group for Child Mortality Estimation (UNICEF, WHO, World Bank, UN DESA Population Division) 2021, under-five mortality was 49 deaths per 1000 live birth in Ethiopia [15].

The geographical distribution of under-five mortality can vary widely, even between neighboring areas that share similar risk profiles [16–20]. It is therefore essential to understand the spatiotemporal distributions of under-five mortality and investigate environmental and climatic risk factors to inform policymakers for targeted interventions to reduce the burden of under-five mortality.

## Methods

### Country context

The study was conducted in Ethiopia, the second-most populous country in Africa with a total population of 115 million in 2020 [21]. The country has a surface area of 1.1 million km$^2$ and has a variety of geographical features with altitudes ranging from 125 m below sea level in Danakil Depression, Afar, to 4620 m above sea level in Ras Dashen, Amhara. Administratively, Ethiopia is divided into—twelve regional states and two administrative cities (first-level), which are further divided into zones (second-level), districts (third-level), and Kebeles (fourth-level).

Ethiopia is among the top five countries that accounted for more than 60% of the world's under-five mortality. Child health is one of the priorities of the Ethiopian government in the healthcare system. Poor access to healthcare is a major problem to provide maternal and child health services in Ethiopia, with more than half of the population walking more than 10 km to get to the nearest health facility [22].

### Data sources

Data on under-five mortality were obtained from the Ethiopian Demographic and Health Surveys (EDHS), conducted in 2000, 2005, 2011, 2016, and 2019. A polygon shapefile for the Ethiopian administrative boundaries was obtained from the Database for Global Administrative Areas (GADM), a free online database [23]. Under-five mortality data were collected through a retrospective birth history of included women in each EDHS. The live births that occurred five years before the interview were included in each of the five EDHS [24].

We included several ecological-level variables related to child mortality. We chose variables relevant to child survival based on a review of relevant literature [25–28]. The covariates were selected based on the biological plausibility and availability of nationwide data. Climatic variables such as mean annual temperature and mean annual precipitation were obtained from the WorldClim website [29]. Altitude data were obtained from the Shuttle Radar Topography Mission (SRTM) [30]. Data on access to the nearest cities and access to a healthcare facility (i.e., hospital or clinic) were obtained from the Malaria Atlas Project (MAP) [31]. Distance to the water body and population density were extracted from Global Lakes and Wetlands Database (GLWD) and WorldPop, respectively [32].

**Table 1. Covariate correlation result of variables included in this study.**

| Covariates | Access to the health facility | Population density | Temperature | Precipitation | Access to city | Altitude | Distance to the water body |
|---|---|---|---|---|---|---|---|
| Access health facility | 1 | -0.19213 | 0.502995 | -0.52677 | 0.634061 | -0.52212 | -0.0721 |
| Population density | -0.19213 | 1 | -0.23901 | 0.178868 | -0.22035 | 0.244469 | 0.029721 |
| Temperature | 0.502995 | -0.23901 | 1 | -0.64523 | 0.450756 | -0.97829 | -0.19911 |
| Precipitation | -0.52677 | 0.178868 | -0.64523 | 1 | -0.39522 | 0.645797 | 0.168528 |
| Access City | 0.634061 | -0.22035 | 0.450756 | -0.39522 | 1 | -0.45208 | -0.04044 |
| Altitude | -0.52212 | 0.244469 | -0.97829 | 0.645797 | -0.45208 | 1 | 0.20824 |
| Distance to a water body | -0.0721 | 0.029721 | -0.19911 | 0.168528 | -0.04044 | 0.20824 | 1 |

## Spatial analysis

Bayesian model-based geostatistics was used to create a predictive map of under-five mortality over the study area. It was also used to identify factors associated with under-five mortality. The predicted maps of under-five mortality were produced at the lower resolution of 1 km$^2$. The model used in this study is described below. Ecological level risk factors such as mean annual temperature, mean annual precipitations, distance to the water body, access to the health facility, distance to the nearest cities, and population density were fitted with the under-five mortality data at each of the five EDHS into a spatial binomial regression model: $Y_j \sim Binomial(n_j, p_j)$;

$$logit(p_j) = \alpha + \sum_{z=1}^{z} \beta_z \boldsymbol{X}_{z,j} + \zeta_j;$$

where $Y_j$ is the under-five mortality, $n_j$ is the total number of children and $p_j$ is the predicted under-five mortality at location $j$, $\alpha$ is the intercept, $\beta$ is a matrix of covariate coefficients, $\boldsymbol{X}$ is a design matrix of $z$ covariates, and $\zeta_j$ are spatial random effects modeled using a zero-mean Gaussian Markov random field (GMRF) [33, 34]. Parameters were estimated using the Integrated Nested Laplace Approximation (INLA) approach in R (R-INLA) (33, 34).

The covariate correlation matrix was checked and variables, which had interactions were excluded from the model (Table 1).

Models were validated using the conditional predictive ordinates (CPO) and the probability integral transform (PIT) statistics [35, 36]. Both CPO and PIT were obtained as "leave-one-out" cross-validation in INLA. The Watanabe Applicable Information Criterion (WAIC) statistic was used to select the best-fitting model (S1 Table).

## Results

### National and regional under-five mortality rate

A total of 48,750 study participants were included in this analysis. Table 2 shows the national and regional under-five mortality rate in Ethiopia. The national under-five mortality rate in Ethiopia declined from 121 per 1000 live births in 2000 to 59 per 1000 live births in 2019. Spatial variation in under-five mortality was observed at the regional level with the highest rates reported in Afar (155 per 1000 live births), Gambela (135 per 1000 live births), and Amhara (130 per 1000 live births) in 2000 EDHS. Similarly, in 2005 EDHS high mortality rate was observed in Amhara (116 per 1000 live births), Southern, Nations, the Nationalities and Peoples region (94 per 1000 live births), Benshangul-Gumuz (92 per 1000 live births), and Afar (92 per 1000 live births) regions. A high mortality rate was observed in Afar, Somali, Gambela, and Benshangul-Gumuz in 2011, 2016, and 2019 EDHS. In all five surveys, the lowest under-five mortality was observed in Addis Ababa (Table 2).

**Table 2. National and regional under-five mortality rate per 1000 live births in Ethiopia (2000, 2005, 2011, 2016, and 2019).**

| Regions | U5M rate | U5M rate | U5M rate | U5M rate | U5M rate |
|---|---|---|---|---|---|
| | 2000 | 2005 | 2011 | 2016 | 2019 |
| Tigray | 101 | 66 | 66 | 40 | 31 |
| Afar | 155 | 92 | 86 | 85 | 54 |
| Amhara | 130 | 116 | 70 | 50 | 45 |
| Oromia | 129 | 87 | 70 | 55 | 56 |
| Somali | 113 | 89 | 74 | 68 | 89 |
| Benishangul | 114 | 92 | 93 | 73 | 86 |
| SNNPR | 118 | 94 | 76 | 56 | 41 |
| Gambela | 135 | 68 | 81 | 62 | 79 |
| Harari | 98 | 62 | 65 | 68 | 65 |
| Addis Ababa | 88 | 53 | 35 | 30 | 21 |
| Dire Dawa | 129 | 75 | 50 | 57 | 67 |
| Ethiopia | 121 | 87 | 73 | 60 | 59 |

N.B: SNNPR = Southern Nations, Nationalities and peoples Region

## Trends of under-five mortality rate

Fig 1 shows the temporal trends of the under-five mortality rate in Ethiopia from 2000 to 2019. The mortality rate declined in the last two decades from 121 per 1000 live births in 2000 to 59 per 1000 live births in 2019. While the decline in under-five mortality was faster in the first three surveys, it remains stagnant in the last two surveys (Fig 1).

## Spatial clusters of under-five mortality

Spatial variations of under-five mortality were observed at local levels with the highest rates reported in Western, Northeastern, Eastern, and Northcentral parts of Ethiopia. The lowest

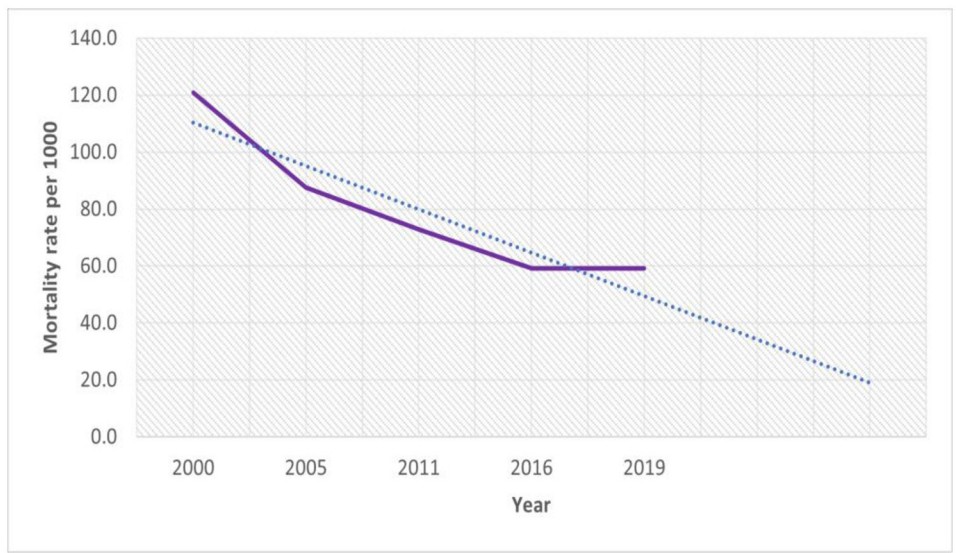

**Fig 1. Trends of under-five mortality rate in Ethiopia from 2000–2019.**

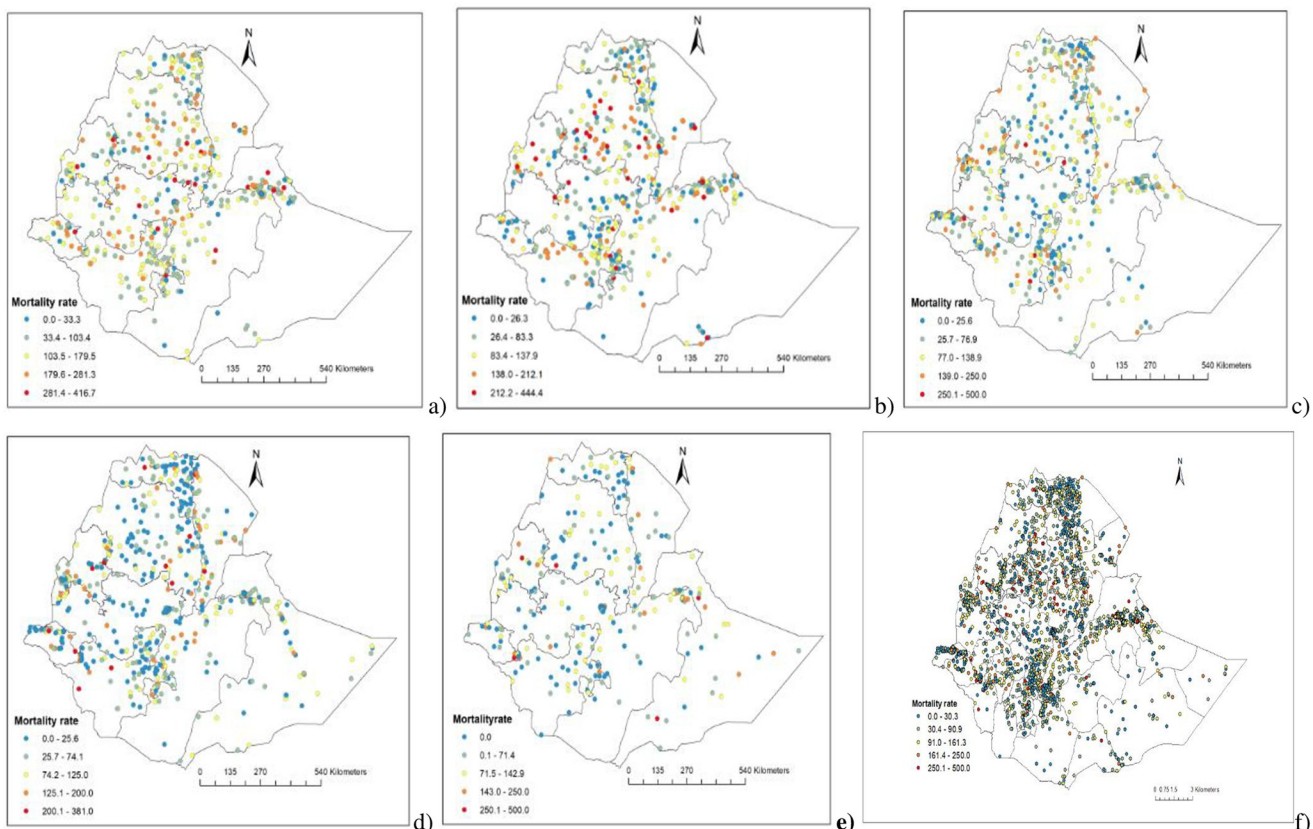

**Fig 2. Geographical locations of data points and under-five mortality in Ethiopia: 2000(a), 2005(b), 2011(c), 2016(d), 2019(e) and 2000–2019 (f).**

rates of under-five mortality were reported in the Central, Northern and Southern parts of Ethiopia (Figs 2 and 3, S1 & S2 Figs).

## Drivers of under-five mortality

Spatial clustering of under-five mortality was significantly associated with population density, access to the water body, and climatic factors such as temperature. The population density was negatively associated with under-five mortality (mean regression coefficient (β): -0.007; 95% credible interval (95% CrI): -0.012, -0.003). Temperature (mean regression coefficient (β): 0.127; 95% credible interval (95% CrI): 0.032, 223) and distance to a water body (mean regression coefficient (β): 0.192; 95% credible interval (95% CrI): 0.051, 0.332) were positively associated with under-five mortality (Table 3).

## Discussion

This study provides spatiotemporal estimates of the under-five mortality rate at national, subnational, and local levels in Ethiopia for 20 years period, from 2000 to 2019. The under-five mortality rate was significantly reduced from 121 1000 live births in 2000 to 59 per 1000 live births in 2019 during the last two decades in Ethiopia. The recent mortality rate in Ethiopia was lower than the under-five mortality rate in Africa (74 per 1000 live births). However, it is higher than the overall global under-five mortality rate (38 per 1000 live births) and the SDG targets (25 death per 1000 live births).

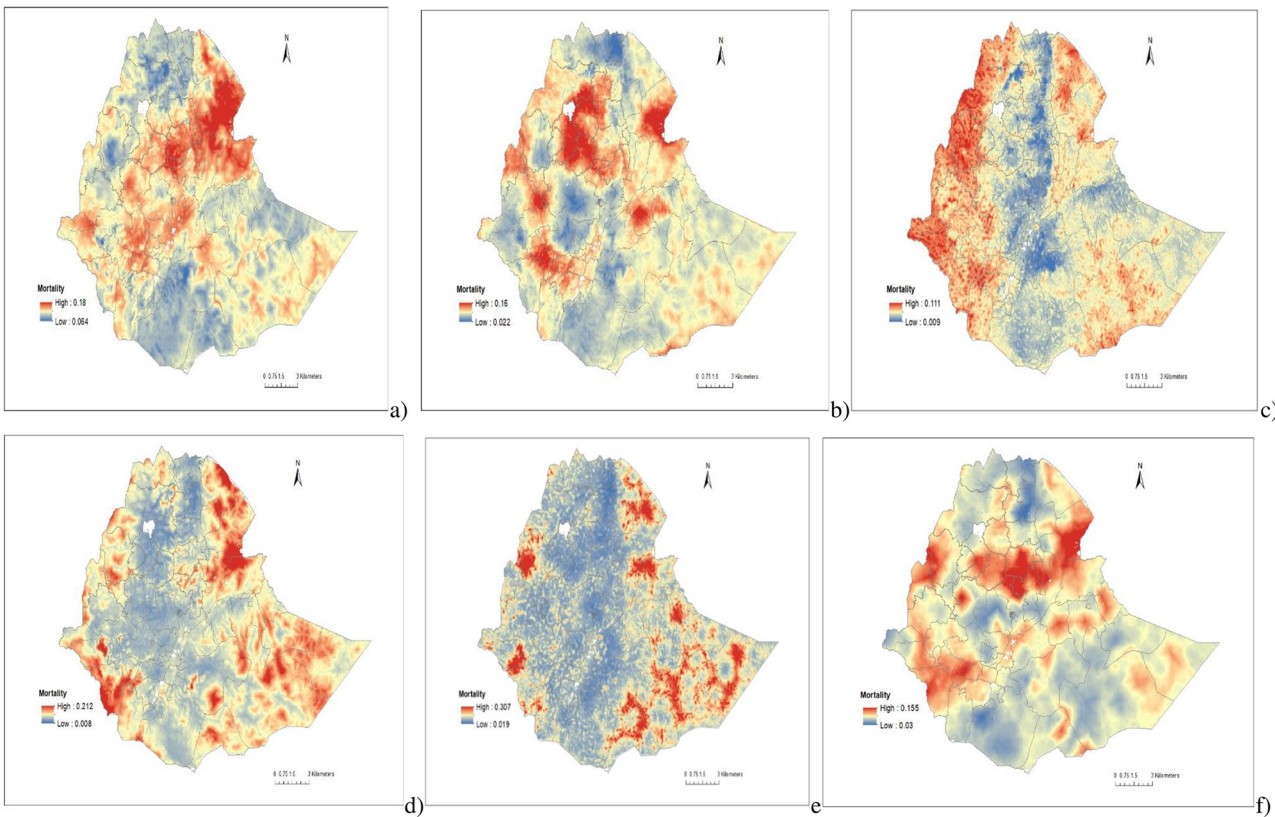

**Fig 3. The predicted geospatial map for under-five mortality in Ethiopia: 2000(a), 2005(b), 2011(c), 2016(d), 2019(e) and 2000–2019(f).**

The under-five mortality rate is an important indicator of a country's health status. The decile in the under-five mortality rate in our study is consistent with a previous study conducted in Northern Ethiopia [37]. The observed decline in the mortality rate in Ethiopia could be due to the increased coverage and quality of antenatal care [38], skilled delivery at birth [39], and postnatal care to the mother and their baby [40]. The improved nutritional supplement and treatment may also contribute to the decline in child mortality in Ethiopia. A recent

**Table 3. Regression coefficient mean and 95% credible intervals (CrI) of covariates included in a Bayesian spatial model with binomial response for the under-five mortality in Ethiopia between 2000 and 2019.**

| Covariates | Under-five mortality Regression coefficients of Mean (95% CrI) | | | | | |
|---|---|---|---|---|---|---|
| | 2000 | 2005 | 2011 | 2016 | 2019 | 2000–2019 |
| Intercept | -2.02(-2.39, -1.73) | -2.38(-2.65, -2.14) | -2.51(-2.64, -2.38) | -2.66(-2.89, -2.44) | -2.78(-3.04, -2.53) | -2.42 (-2.533, -2.314) |
| Access to health facilities | 0.02(-0.24, 0.28) | -0.06(-0.34, 0.22) | -0.05(-0.25, 0.15) | 0.05(-0.21, 0.30) | 0.27(-0.01, 0.55) | 0.033 (-0.076, 0.14) |
| Population density | **-0.01(-0.012, 0.001)** | **-0.02(-0.03, -0.004)** | **-0.02(-0.03, -0.005)** | -0.006(-0.02, 0.004) | 0(-0.02, 0.02) | -0.007 (-0.012, -0.003) |
| Temperature | 0.07(-0.03, 0.17) | 0.11(-0.02, 0.25) | **0.13(0.03, 0.22)** | 0.12(-0.03, 0.28) | 0.15(-0.07, 0.36) | 0.043 (-0.027, 0.113) |
| Precipitation | -0.001(-0.14, 0.13) | 0.05(-0.13, 0.23) | 0.08(-0.02, 0.17) | 0.05(-0.13, 0.25) | 0.03(-0.22, 0.26) | 0.037 (-0.061, 0.129) |
| Access to cities | -0.08(-0.21, 0.05) | -0.03(-0.22, 0.15) | 0.04(-0.10, 0.17) | 0.17(-0.01, 0.35) | -0.05(-0.33, 0.22) | -0.028 (-0.106, 0.049) |
| Distance to a water body | -0.003(-0.07, 0.06) | -0.02(-0.09, 0.06) | 0.04(-0.03, 0.11) | -0.03(-0.12, 0.07) | **0.19(0.05, 0.33)** | 0.003 (-0.033, 0.038) |

CrI: credible interval; bold fonts show 'statistically significant' results within a Bayesian framework (no zero within the 95% CrI).

study showed that nutritional supplements contributed to a 45% reduction in under-five mortality [3]. Increased health facility delivery was another important factor that reduce 80% of under-five mortality [5]. Additionally, the Ethiopian government had a strong commitment to improving health access through training a large number of health extension workers and expansion of health posts in the country, which may also be contributed for the decline in child mortality in the last two decades.

Consistent with the previous studies conducted in Ethiopia, the spatial clustering of under-five mortality was observed in Somali, Benshangul-Gumuz, and Gambela during the five surveys [41]. This might be related to health care access, which affects antenatal care utilization, institutional delivery, and postnatal care utilization, which increases the probability of child death. These three regions are where nomadic and pastoralists lived, in these regions access to a safe and adequate water supply is difficult. They are also highly endemic areas for malaria. In addition, childhood immunization coverage in these regions is very low [42]. Maternal and child health service utilization are also very poor in these parts of the country, which may contribute to the observed high child mortality. Strengthening community-based health extension programs would be essential in these areas. Moreover, providing training for health professionals and providing maternal health education for child health including infant feeding practice and child care utilization may improve child survival in these regions [43, 44].

The population density was negatively associated with under-five mortality. This means that there was low child mortality in highly populated areas (i.e., urban areas). This might be because healthcare access, maternal education, adequate water source, and child nutritional practice are relatively good in urban areas [45].

In agreement with a previous study conducted in Burkina Faso [46], the temperature was positively associated with under-five mortality. This might be because in the area with high temperatures the risk of malaria transmission [47] and diarrheal diseases [48] was high, which are the major killers of under-five children [49].

In addition, distance to a water body was positively associated with under-five mortality. This might be related to sanitation and toilet facilities that increase the risk of diarrheal disease and child death [50]. The walking time to the main water source of the household is strongly associated with under-five death [51]. Improving access to safe and adequate water sources all over the country would be essential to reducing childhood mortality.

This study has paramount importance to identify risk areas of under-five mortality in Ethiopia, which may inform policymakers to provide area-specific interventions. The study was using two-decade nationwide data that could provide a reliable estimate of under-five mortality over time and the change in the two decades. However, the study has some limitations. First, it did not include some important covariates such as agricultural productivity, malaria prevalence, community literacy, and poverty level that could influence under-five mortality due to a lack of data. Second, there are ecological fallacies that again could influence under-five mortality, since we collected the covariates at different sources with different resolutions.

## Conclusion

In Ethiopia, under-five mortality declined over the last two decades but still, it is unacceptably high with substantial variations at sub-national and local levels. The spatial clustering of under-five mortality was observed in Western, Northeastern, Eastern, and Northcentral parts of Ethiopia. The population density was negatively associated with under-five mortality in Ethiopia. Therefore, interventions targeted to reduce under-five mortality should be strengthened in the areas that had a clustering of under-five mortality in Ethiopia by increasing access to quality health care access.

## Supporting information

**S1 Table. Watanabe-Akaike information criterion (WAIC) values corresponding to different model specifications in all the five EDHS (2000–2019).**
(DOCX)

**S1 Fig. Geographical locations of data points and under-five mortality in Ethiopia: 2000 (A), 2005(B), 2011(C), 2016(D), 2019(E).**
(DOCX)

**S2 Fig. The predicted geospatial map for under-five mortality in Ethiopia: 2000(A), 2005 (B), 2011(C), 2016(D), and 2019(E).**
(DOCX)

## Author Contributions

**Conceptualization:** Kendalem Asmare Atalell, Kefyalew Addis Alene.

**Data curation:** Kendalem Asmare Atalell, Kefyalew Addis Alene.

**Formal analysis:** Kendalem Asmare Atalell, Kefyalew Addis Alene.

**Funding acquisition:** Kendalem Asmare Atalell.

**Investigation:** Kendalem Asmare Atalell, Kefyalew Addis Alene.

**Methodology:** Kendalem Asmare Atalell, Kefyalew Addis Alene.

**Project administration:** Kendalem Asmare Atalell.

**Resources:** Kendalem Asmare Atalell.

**Software:** Kendalem Asmare Atalell, Kefyalew Addis Alene.

**Supervision:** Kendalem Asmare Atalell, Kefyalew Addis Alene.

**Validation:** Kendalem Asmare Atalell, Kefyalew Addis Alene.

**Visualization:** Kendalem Asmare Atalell, Kefyalew Addis Alene.

**Writing – original draft:** Kendalem Asmare Atalell.

**Writing – review & editing:** Kendalem Asmare Atalell, Kefyalew Addis Alene.

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
