## [Decision Letter · Decision Letter 0]

4 Nov 2022

PGPH-D-22-01440

Spatiotemporal distributions of under-five mortality in Ethiopia between 2000 and 2019

Dear Dr. Atalell,

Thank you for submitting your manuscript to PLOS Global Public Health. After careful consideration, we feel that it has merit but does not fully meet PLOS Global Public Health’s publication criteria as it currently stands. Therefore, we invite you to submit a revised version of the manuscript that addresses the points raised during the review process.

We look forward to receiving your revised manuscript.

Kind regards,

Prevost Jantchou

Academic Editor

Journal Requirements:

2. Please provide separate figure files in .tif or .eps format only and remove any figures embedded in your manuscript file. Please also ensure that all files are under our size limit of 10MB.

Additional Editor Comments (if provided):

Thanks for submitting the work to PLOS GLOBAL PUBLIC HEALTH. The reviewers have completed the review of your manuscripts and here are the comments that need to be addresssed.

Reviewer 1

1/ This study is very welcome to provide useful data on under 5 mortality in Ethiopia and then focus healt care interventions in appropriated areas.

2/ It would have been useful to break down the data on mortality according to the age of the children because mortality causes at ages 6 months and 4 years could be very different. Do you have such data?

3/ The main limitation to publishing this paper at this stage is the English. Reviewing the global wording of the manuscript would be mandatory. See below some examples:

line 61-62 : wording is not correct

line 87 : children were participated

Line 111 : was substantially declined

Reviewer 2

Congratulations on your paper! It is a fine piece of work and adds relevant information about under-five mortality.

There are some comments to share:

Background: concise, straightfoward, shows essential data.

Methods:

It is very important to contextualize readers about the country reality.

Data was obtained from different sources, what can compromise your results. However, the model you have used explains your argument.

Results:

Great figures! Organized tables.

There a typo mistake at Line 87: probably you mean Table 2, not Table 1, as you mention at the end of the paragraph.

Discussion:

Well designed, relevant information.

In the Result section, you present data by region (Western, Northeastern, Eastern, and Northcentral). Here you cite specific districts. It is difficult to relate to the figures if there is no mark on the map. By combining both information, citing region and district, readers could better understand you data.

Comparing Map A and Map E on Figure 3, there is a decline on mortality during the years. Did you find any reason for that? It would be very interesting to understand what happened.

Conclusion:

Concise and clear.

Reviewers' comments:

Reviewer's Responses to Questions

**Comments to the Author**

1. Does this manuscript meet PLOS Global Public Health’s publication criteria? Is the manuscript technically sound, and do the data support the conclusions? The manuscript must describe methodologically and ethically rigorous research with conclusions that are appropriately drawn based on the data presented.

Reviewer #1: Yes

Reviewer #2: Yes

2. Has the statistical analysis been performed appropriately and rigorously?

Reviewer #1: Yes

Reviewer #2: I don't know

3. Have the authors made all data underlying the findings in their manuscript fully available (please refer to the Data Availability Statement at the start of the manuscript PDF file)?

Reviewer #1: Yes

Reviewer #2: Yes

4. Is the manuscript presented in an intelligible fashion and written in standard English?

Reviewer #1: Yes

Reviewer #2: No

5. Review Comments to the Author

Reviewer #1: Congratulations on your paper! It is a fine piece of work and adds relevant information about under-five mortality.

There are some comments to share:

Background: concise, straightfoward, shows essential data.

Methods:

It is very important to contextualize readers about the country reality.

Data was obtained from different sources, what can compromise your results. However, the model you have used explains your argument.

Results:

Great figures! Organized tables.

There a typo mistake at Line 87: probably you mean Table 2, not Table 1, as you mention at the end of the paragraph.

Discussion:

Well designed, relevant information.

In the Result section, you present data by region (Western, Northeastern, Eastern, and Northcentral). Here you cite specific districts. It is difficult to relate to the figures if there is no mark on the map. By combining both information, citing region and district, readers could better understand you data.

Comparing Map A and Map E on Figure 3, there is a decline on mortality during the years. Did you find any reason for that? It would be very interesting to understand what happened.

Conclusion:

Concise and clear.

Reviewer #2: 1/ This study is very welcome to provide useful data on under 5 mortality in Ethiopia and then focus healt care interventions in appropriated areas.

2/ It would have been useful to have the data of mortality according to the age of the chidren because mortality causes at age 6 months and 4 years could be very different. Do you have such data?

3/ The main limitation to publish this paper at this stage is the English. Reviewing the global wording of the manuscriptip would be madatory. See below some examples :

line 61-62 : wording is not correct

line 87 : children were participated

Line 111 : was substantially declined

4/ I'm not able to have a critical view on statistics because of lack of skill

6. PLOS authors have the option to publish the peer review history of their article (what does this mean?). If published, this will include your full peer review and any attached files.

**Do you want your identity to be public for this peer review?** For information about this choice, including consent withdrawal, please see our Privacy Policy.

Reviewer #1: **Yes: **Roberta Maria P Azevedo

Reviewer #2: No

---

## [Decision Letter · Decision Letter 1]

21 Feb 2023

Spatiotemporal distributions of under-five mortality in Ethiopia between 2000 and 2019

PGPH-D-22-01440R1

Dear Dr. Atalell,

We are pleased to inform you that your manuscript 'Spatiotemporal distributions of under-five mortality in Ethiopia between 2000 and 2019' has been provisionally accepted for publication in PLOS Global Public Health.

Best regards,

Prevost Jantchou

Academic Editor

Reviewer Comments (if any, and for reference):

Reviewer's Responses to Questions

**Comments to the Author**

1. If the authors have adequately addressed your comments raised in a previous round of review and you feel that this manuscript is now acceptable for publication, you may indicate that here to bypass the “Comments to the Author” section, enter your conflict of interest statement in the “Confidential to Editor” section, and submit your "Accept" recommendation.

Reviewer #1: All comments have been addressed

Reviewer #2: All comments have been addressed

2. Does this manuscript meet PLOS Global Public Health’s publication criteria? Is the manuscript technically sound, and do the data support the conclusions? The manuscript must describe methodologically and ethically rigorous research with conclusions that are appropriately drawn based on the data presented.

Reviewer #1: Yes

Reviewer #2: Yes

3. Has the statistical analysis been performed appropriately and rigorously?

Reviewer #1: Yes

Reviewer #2: I don't know

4. Have the authors made all data underlying the findings in their manuscript fully available (please refer to the Data Availability Statement at the start of the manuscript PDF file)?

Reviewer #1: Yes

Reviewer #2: Yes

5. Is the manuscript presented in an intelligible fashion and written in standard English?

Reviewer #1: Yes

Reviewer #2: Yes

6. Review Comments to the Author

Reviewer #1: Congratulations on your paper! You have done great changes, mostly concerning language issues and now your text is more cohesive and reliable.

After your study, more health and social care interventions can be properly applied, providing better living conditions to this population. Since it was possible to reduce mortality during the study years, more public policies can be structured to the main issues of the community.

Reviewer #2: (No Response)

7. PLOS authors have the option to publish the peer review history of their article (what does this mean?). If published, this will include your full peer review and any attached files.

**Do you want your identity to be public for this peer review?** For information about this choice, including consent withdrawal, please see our Privacy Policy.

Reviewer #1: **Yes: **Roberta Maria P Azevedo

Reviewer #2: No
